A 22-amino-acid peptide regulates tight junctions through occludin and cell apoptosis

Zhu Maoying 1
Lu Juan 1
Shen Jianyun 1
Fei Lumin 1
Chen Deyu chendeyu7104@aliyun.com 2
1 College of Biological and Food Engineering, Fuyang Normal University , Fuyang , Anhui , China
2 College of Medicine, Fuyang Normal University , Fuyang , Anhui , China
Gillespie Joseph
Electronic publication date: 2020 Nov 3
Publication date: 2020
Volume: 8
Electronic Location ID: e10147
Received 2020 Jul 22; Accepted 2020 Sep 21
Copyright: ©2020 Zhu et al.
Copyright year: 2020
Copyright holder: Zhu et al.
License: This is an open access article distributed under the terms of the Creative Commons Attribution License, which permits unrestricted use, distribution, reproduction and adaptation in any medium and for any purpose provided that it is properly attributed. For attribution, the original author(s), title, publication source (PeerJ) and either DOI or URL of the article must be cited.
License URL: https://creativecommons.org/licenses/by/4.0/

Keywords: Occludin, TJ, Apoptosis, Sertoli cell, Spermatid

Funding: National Natural Science Fund 81771567 Anhui Province Higher School Natural Science Key Program KJ2017A338 This study was sponsored by the National Natural Science Fund (81771567) and the Anhui Province Higher School Natural Science Key Program (KJ2017A338). The funders had no role in study design, data collection and analysis, decision to publish, or preparation of the manuscript.

==============================
Occludin is a structural protein of tight junctions (TJ) in the blood–testis barrier (BTB). A 22-amino-acid peptide (22AA) in the second extracellular loop can reversibly regulate TJ, but its regulatory mechanism is unknown. In this study, a 22AA-induced TJ destruction animal model was constructed to investigate the effect of 22AA on Sertoli cells (SCs) and spermatid counts and cell apoptosis at different time points using a multiplex immunofluorescence technique. The effect of 22AA on the location and distribution of occludin was analyzed via dual confocal fluorescence microscope. Western blotting was used to analyze dynamic changes in occludin expression. Real-time RT-PCR was used to analyze miR-122-5p expression changes. Sperm density counts and mating methods were used to analyze the effect of 22AA on fertility in mice. The results showed that 22AA promoted SC and spermatid apoptosis, downregulated occludin, upregulated miR-122-5p, and decreased sperm density and litter size before 27 days (27D). After 27D, the expression of occludin increased again, miR-122-5p expression decreased again, both sperm density and litter size returned to normal, apoptosis stopped, and spermatogenesis began to recover. Therefore, it can be concluded that 22AA can destroy TJ by downregulating occludin and inducing cell apoptosis. After 27D, TJ and spermatogenesis functions return to normal.

Introduction

In the testis, there is a blood–testis barrier (BTB) between the seminiferous tubules and blood vessels. The BTB composition includes the interstitial capillary endothelium and its basement membrane, connective tissue, and tight junctions (TJ) between the basement membrane of the seminiferous epithelium and Sertoli cells (SCs). TJ are the main structures that constitute the BTB (Chi et al., 2017). One cause of male infertility is TJ abnormality in SCs, resulting in blockage of spermatogenic cell migration in seminiferous tubules (Lui, Lee & Cheng, 2001). The three-dimensional configuration of SCs is complex. SCs are irregularly cone-shaped, with the base closely touching the basement membrane, the top extending into the lumen, and many irregular depressions in the lateral surface and lumen surface, in which various levels of spermatogenic cells are embedded. Membranes near the basal side of the adjacent SCs form TJ, and the seminiferous epithelium is divided into a basal compartment and adluminal compartment (Li et al., 2016). The basal compartment is located between the basement membrane of seminiferous epithelium and the TJ of SCs and contains spermatogonia cells (including type A and type B). The adluminal compartment, located above the TJ, connects with the lumen of seminiferous tubules and contains spermatocytes, germ cells, and sperm (Tanaka et al., 2019). During spermatogenesis, the spermatocytes at the preleptotene stage and the leptotene stage differentiated from type B spermatogonia cells must move across the TJ into the abluminal compartment to complete their development (Ramos-Trevino et al., 2017). Therefore, the movement of germ cells across the seminiferous epithelium during spermatogenesis is closely related to structural reconstruction of TJ.

Many TJ experimental models exist for studying the BTB. The in vitro BTB experimental model uses the testicular SC primary dual-chamber culture method to determine TJ tightness by measuring transepithelial electrical resistance (Feng et al., 2019). Glycerol (Wiebe et al., 2000), CdCl2 (Hew et al., 1993), and blocking peptides (Chung et al., 2001) can also destroy testicular TJ. However, the destruction of testicular TJ induced by glycerin injection or CdCl2 injection is irreversible. Thus, studying the dynamic changes in TJ-related molecules during the disintegration process and reconstruction process using these two animal models is difficult.

Structural proteins of TJ include occludin (Liu et al., 2018), zonula occludens-1 (ZO-1) (Fanning et al., 1998), ZO-2 (Gonzalez-Mariscal et al., 2000), ZO-3, the claudin multigene family, and adhesion molecules (Furuse et al., 1998). Occludin is a 65-kDa protein localized at TJ  (McCarthy et al., 1996; Du et al., 2010) that consists of four transmembrane domains, a long carboxyl-terminal cytoplasmic domain, a short N-terminal cytoplasmic domain, two extracellular loops, and one intracellular loop (Ghassemifar et al., 2002). Its structure is highly conserved among different mammalian species (Wong et al., 2007). The first extracellular loop is rich in tyrosine and glycine, accounting for approximately 60% of the amino acid residues, and is involved in intercellular adhesion (Bamforth et al., 1999). The second extracellular loop is involved TJ (Medina et al., 2000). Occludin has also been reported to induce apoptosis and apoptotic sensitization, which are regarded as antitumorigenic activities (Gu et al., 2008). However, the mechanism underlying the specific role of occludin in cell apoptosis remains poorly understood.

Wong & Gumbiner (1997) used an in vitro cell model to add an artificially synthesized 44-amino-acid short peptide identical to the second extracellular loop sequence of occludin into the Xenopus kidney epithelial cell line A6, thereby reducing the tightness of TJ. Chung et al. (2001) confirmed that a 22-amino-acid peptide (22AA) in the second extracellular loop of occludin could reversibly regulate TJ. However, Chung et al. (2001) only analyzed the effect of 22AA on the morphological structure of the seminiferous tubules. What is the mechanism underlying the regulation of TJ by 22AA? Is the expression and localization of occludin affected by 22AA? How are SC and spermatid counts in the seminiferous tubules dynamically affected by 22AA? Can 22AA affect the number of offspring? The above questions should be studied in depth. To further investigate the mechanism underlying the regulation of TJ by 22AA, this study analyzed the effect of 22AA on occludin expression and localization, SC and spermatid apoptosis, and mouse fertility using a TJ damage animal model.

Experimental Materials and Methods

Experimental animals

Specific-pathogen-free Kunming (KM) mice, 6 to 8 weeks old, with a body weight of 30–35 g, were provided by Chongqing Enswell Biotechnology Co., Ltd., China. Animals were kept at a temperature of 23−25 °C with a 12 h light:12 h dark (12 h–12 h) cycle. The animals were allowed food and water ad libitum. The experiment was approved by the Experimental Animal Ethics Committee of Fuyang Normal University, China (Grant No. 20200006).

Construction of animal models

The sequence of 22AA is NH2-GSQIYTICSQFYTPGGTGLYVD-COOH (from the 209th to 230th amino acid of occludin) (Chung et al., 2001), and 22AA was synthesized by Beijing Protein Innovation Co., Ltd., China. To prepare a 200.0 g/L solution, 22AA was dissolved in 0.9% sterile saline. Male KM mice were anesthetized with 7% chloral hydrate (0.5 mL/100 g), the scrotal skin was cut, both (left and right) testicles were exposed, and a 26-gauge needle (Becton Dickinson, Rutherford, NJ) was used for injection of 7.0 µL of 22AA solution at three different locations in each testicle. The control group was injected with 0.9% sterile saline. After injection, the testicles were sutured, and the mice were put back into the cage. Thirty-six experimental animals were divided into six groups, with six mice in each group: A, normal control group; B, 7 days (7D) after injection of 22AA; C, 17D after 22AA injection; D, 27D after 22AA injection; E, 37D after 22AA injection; and F, 47D after 22AA injection. After the mice were anesthetized by intraperitoneal injection of 7% chloral hydrate (0.5 mL/100 g) at different time points, the skin of the scrotum was removed, and one testis was removed and stored at −80 °C for western blotting (WB) and quantitative PCR (qPCR). The other testis was fixed with 4% paraformaldehyde for morphological examination. At the end of the experiment, the mice were euthanized using 0.3 mL 7% chloral hydrate (0.5 mL/100 g) according to the American Veterinary Medical Association (AVMA) Guidelines for the Euthanasia of Animals, 2013 Edition.

Multiplex immunofluorescence detection

Testicular tissues were collected at different stages, fixed and blocked using conventional methods, embedded in paraffin, deparaffinized, dehydrated, and subjected to heat-induced antigen retrieval. The types and sources of antibodies used in the experiments are shown in Table 1. The sections were washed with distilled water, immersed in phosphate-buffered saline (PBS) for 5 min, and then blocked with goat serum at room temperature for 60 min. The blocking solutions were then aspirated, primary anti-Bax antibody (1:200) was added dropwise, and the sections were placed in a humidified box and incubated at 4 °C overnight. The next day, the sections were incubated at room temperature for 30 min. The sections were immersed and washed in PBS three times for 3 min each. After the sections were blotted, a fluorescent Cy3-labeled secondary antibody (1:800 dilution) was added dropwise, followed by incubation at 37 °C for 60 min in a humidified box. The sections were washed three times with PBS for 3 min each. After the sections were blotted, primary anti-WT1 or anti-Prm2 antibody (1:100 dilution) was added dropwise, and the sections were incubated in a humidified box at 37 °C for 60 min. The sections were washed three times with PBS for 3 min each. After the sections were blotted, a fluorescent FITC-labeled secondary antibody (1:800) was added dropwise, followed by incubation at 37 °C for 60 min in a humidified box. After three washes with PBS for 3 min each, DAPI was added dropwise, followed by incubation in the dark for 15 min for nuclear staining. Then, the sections were washed for 5 min four times. After excess DAPI was washed away, the slides were mounted in anti-fluorescence quencher mounting medium. The sections were then observed under a confocal fluorescence microscope (Leica, Germany), and images were collected. Fifty sections in different parts of each testicle were examined. Images were quantitatively analyzed using ImageJ software. The relative expression value of Bax was determined by the optical density value.

Table 1 Source of reagents in this study.

Types and sources of antibodies	Origin	
Primary antibody: anti-Bax (rat origin)	Abcam, USA	
Primary antibody: anti-WT1 (rabbit origin)	Abcam, USA	
Primary antibody: anti-Prm2 (rabbit origin)	Proteintech, USA	
Secondary antibodies: anti-Alexa Fluor CY3 (anti-mouse)	Abcam, USA	
Secondary antibody: anti-Alexa Fluor 488 (anti-rabbit)	Abcam, USA	
Immunofluorescence primary antibody (diluted)	Beyotime, China	
Immunofluorescence secondary antibody (diluted)	Beyotime, China	
PBS	ZSGB-BIO, Chia	
Paraformaldehyde	Shanghai Sangong	
Goat serum	HyClone, USA	
DAPI staining solution	Beyotime, China	
Anti-fluorescence quenching agent	Beyotime, China	
Primary antibody: occludin (rabbit origin)	Abcam, USA	
Secondary antibody: Anti-Alexa Fluor cy3 (anti-rabbit)	Abcam, USA	
Immunofluorescence primary antibody (diluted)	Beyotime, China	
Immunofluorescence secondary antibody (diluted)	Beyotime, China	
PBS	ZSGB-BIO, Chia	
Paraformaldehyde	Shanghai Sangong	
Goat serum	HyClone, USA	
DAPI staining solution	Beyotime, China	
Anti-fluorescence quenching agent	Beyotime, China	

Immunofluorescence analysis of occludin localization and distribution

Testicular tissues were collected at different stages, fixed and blocked by conventional methods, embedded in paraffin, deparaffinized, dehydrated, and subjected to heat-induced antigen retrieval. The sections were cooled at room temperature for 10-20 min, rinsed with distilled water, and immersed in PBS for 5 min. Sections received 0.5% Triton X-100 dropwise, followed by incubation at room temperature for 60 min and three PBS washes of 3 min each. The sections were blocked at room temperature for 60 min via dropwise addition of normal goat serum. The blocking solution was aspirated, and primary anti-occludin antibody (Abcam, America) diluted 1:100 was added dropwise. The sections were placed in a humidified box and incubated at 4 °C overnight. The humidified box was removed and rewarmed to room temperature for 30 min. Sections were washed with PBS three times for 3 min each. After the excess liquid was removed from the sections, Alexa Fluor Cy3-labeled antibody (Abcam, America) diluted 1:800 was added dropwise, followed by incubation at 37 °C for 60 min in a humidified box. Sections were washed with PBS three times for 3 min each, and DAPI (Beyotime, China) was added dropwise for nuclear staining in the dark for 15 min. After excess DAPI was washed away, the slides were mounted using anti-fluorescence quencher mounting medium. The sections were then observed under a confocal fluorescence microscope (Leica, Germany), and images were collected. Fifty sections in different parts of each testicle were examined.

Detection of occludin via WB

After extraction of total proteins from testes at different stages, the total protein content was determined using a BCA kit (Solarbio, China). After sodium dodecyl sulfate-polyacrylamide gel electrophoresis (SDS-PAGE) at 100 V for 1.5 h, samples were transferred to a nitrocellulose (NC) membrane, which was blocked for 2 h via gentle shaking in 50.0 mL of TTBS (20.0 mmol/L Tris, pH 7.5, 0. 5 g/L Tween-220, 8.0 g/L NaCl) containing 50 g/L skim milk powder. The NC membrane was placed in a plastic mantle and sealed after addition of 5.0 mL of rabbit anti-mouse occludin monoclonal antibody (Abcam, America) at 1:500 dilution and then incubated at 4 °C overnight. The NC membrane was taken out and washed with TTBS three times for 15 min each. Then, 5.0 mL of horseradish peroxidase-conjugated goat anti-rabbit IgGI (1:2 000 dilution with TBS) (Jackson, 111-035-008) was added and incubated with the membrane with gentle shaking for 2 h. The NC membrane was taken out and washed with TTBS three times for 10 min each. novaECL reagent was added to the front surface of the NC membrane and allowed to stand for 1 min, and then, light-sensitive films were observed in a dark room. The band density was scanned using a digital gel image analysis system, and the gray value for density was measured with Image Lab 4.1. Using β-actin as the internal control, the expression of occludin was determined by the ratio of the gray values of occludin to those of β-actin.

Real-time RT-PCR analysis of miR-122-5p

Testicular tissues were collected at different time points, and RNA was extracted using a Trizol Total RNA Extraction Kit (Shanghai Sangong, China, catalog number: B511321). According to the kit manual, RNA was extracted and reverse-transcribed into cDNA. The primers for miR-122-5p RT-PCR were F: CCTGGAGTGTGACAATG and R: GAGCAGGCTGGAGAA. The primers for the internal control actin were F: GAGACCTTCAACACCCCAGC and R: ATGTCACGCACGATTTCCC. The BR Green I protocol of the SYBR Green I method was used. The real-time fluorescence PCR kit TransStart Green qPCR SuperMix (catalog no. AQ131-01) was used for PCR amplification on a LightCycler 96 system (Roche, US). The reaction system was as follows: 25.0 µL of 2 × PCR buffer, 5.0 µL of primers (25.0 pmol/µL), 0.5 µL of SYBR green I (20 ×), 2.0 µL of template (cDNA), and 21.5 µL of DEPC water. The amplification conditions were as follows: 94 °C for 4 min; 35 cycles of 94 °C for 20 s, 60 °C for 30 s, and 72 °C for 30 s; followed by 72 °C for detection. Relative expression was calculated using the 2−ΔΔCt method.

Analysis of sperm density

Male mice were sacrificed via cervical dislocation at various time points. After abdominal disinfection, the abdominal wall was cut open to expose the reproductive system. The epididymis was separated via aseptic surgery, and the mesentery and fat surrounding the epididymis were removed with ophthalmic scissors and rinsed. The epididymis was shred and placed in a Petri dish containing 37 °C PBS and then incubated at 37 °C with 5% CO2 and saturated humidity for 30 min. The sperm density was calculated using a cell counting plate after the sperm had spontaneously spread out.

Analysis of litter size

Male mice were co-caged with female mice (female:male = 1:1) at each time point. The time when a vaginal plug was detected was taken as the 0 day of pregnancy. On the 14th day of pregnancy, the mice were sacrificed by cervical dislocation, the uterus was removed by laparotomy, and the number of embryos in the bilateral uterus was recorded.

Statistical analysis

The experimental results are expressed as the mean ± standard deviation. GraphPad Prism 6 was used to complete the data processing. Differences between control group and experimental group were examined using a t-test. Differences among the six different experimental groups were examined using two-way ANOVA, and correlations between groups were examined with a Pearson test. The significant difference level was set as P < 0.01.

Figure 1 Effects of 22AA on the apoptosis of SCs, sperm cells and the localization and distribution of occludin.

Image A, G and M were control panels, others image were different treatment panels.Effects of 22AA on the apoptosis of SCs. Many SCs were present in the control group, with clear intercellular boundaries and a small amount of red Bax distribution (A). The morphology and number of SCs at 7D were not significantly different from those of the control group (B). At 17D and 27D, there were no blue nuclei in the seminiferous tubules, and the intercellular boundary completely disappeared (C, D). The WT1 patch was scattered throughout the seminiferous tubules, with no cells present in the tubules. The tubules were filled with a large amount of red Bax. A few SCs began to appear in the basal layer of the seminiferous tubules at 37D (E). At 47D, the morphology, structure and number of SCs in the seminiferous tubules were not different from those in the control group (F). Effect of 22AA on the apoptosis of sperm cells. Nearly 55 ± 5 spermatids were found in the seminiferous tubules of the control group (G), and the expression level of Bax was 11.245 ± 4.868. At 7D, the spermatid count decreased to 15 ± 3 (H), and the expression of Bax increased to 19.569 ± 6.158. No spermatid was found in the seminiferous tubules at 17D (I) or 27D (J), and the expression levels of Bax increased to 23.467 ± 5.327 and 31.353 ± 13.139, respectively. At 37D, a small amount of Prm2 was distributed in the seminiferous tubules (K), and the expression level of Bax decreased to 16.362 ± 3.267. At 47D, 20 ± 4 spermatids were rediscovered in the seminiferous tubules (L), and the expression level of Bax was 10.176 ± 1.682, which was not different from the control level. Effect of 22AA on the localization and distribution of occludin. Occludin was mainly located at TJ between the basement membrane of seminiferous tubules and SCs in the control group (M). At 7D, a small amount of occludin was distributed at TJ between the basement membrane of seminiferous tubule and SCs. However, the total number of cells in seminiferous tubules was less than that in the control group (N). At 17D and 27D, no cells with blue nuclei were found in the seminiferous tubules (O,P). From day 17 to day 27, the expression of occludin gradually decreased. At 37D, occludin expression began to increase and was found to be distributed on the basement membrane of seminiferous tubules and SCs (Q). At 47D, occludin expression and distribution and the morphological structure of seminiferous tubules were highly similar to those in the control group(R). Bar−, 0.2 µm.

Results

22AA promotes SC apoptosis

Multiplex immunocytochemistry was used to analyze the effect of 22AA on SC apoptosis. Bax was used as the cell apoptosis marker protein to calculate the SC apoptosis rate in seminiferous tubules. WT1 was used as a marker of SCs. The multiplex immunohistochemical results for each group are shown in Fig. 1A. Many SCs were present in the seminiferous tubules of the control group, with clear intercellular boundaries and a small amount of red Bax distribution, and the expression level of Bax was relatively low at 11.373 ± 10.532. At 7D, the SC count in the seminiferous tubules was not significantly different from that in the control group, and the expression level of Bax was 16.783 ± 10.157, which was significantly higher than that in the control group (P < 0.01), suggesting that cell apoptosis started in the seminiferous tubules. At 17D and 27D, no cells with blue nuclei were found in the seminiferous tubules, and the intercellular boundary completely disappeared. The WT1 patch was scattered throughout the seminiferous tubules, and no cell structure was found in the tubules. The seminiferous tubules were filled with a large amount of red Bax, and the expression levels of Bax were 20.521 ± 5.781 and 30.253 ± 12.274 at 17D and 27D, respectively, which were significantly higher than those in the control group (P < 0.01), indicating that at this stage all cells in the seminiferous tubules were already apoptotic. At 37D, the expression level of Bax was 20.862 ± 3.243, which was roughly equivalent to the expression level of Bax at 17D, indicating that apoptosis had begun to terminate; a few SCs began to appear in the basal layer of seminiferous tubules; and TJ reconstruction started. At 47D, the number of SCs and the Bax expression level were not significantly different from those in the control group. These results demonstrate that 22AA promoted SC apoptosis.

22AA reversibly regulates spermatid apoptosis

Multiplex immunofluorescence was used to analyze the effect of 22AA on spermatids. Bax was used as the apoptosis marker protein to analyze spermatid apoptosis in the seminiferous tubules. Prm2 was used as the spermatid marker to determine the spermatid count. The multiplex immunohistochemical results for each group are shown in Fig. 1B. Nearly 55 ± 5 spermatids were found in the seminiferous tubules of the control group, and the expression level of Bax was 11.245 ± 4.868. At 7D, the spermatid count decreased to 15 ± 3, and the expression of Bax increased to 19.569 ± 6.158, suggesting gradual spermatid apoptosis. No spermatid was found in the seminiferous tubules at 17D or 27D, and the expression levels of Bax increased to 23.467 ± 5.327 and 31.353 ± 13.139, respectively, indicating complete spermatid apoptosis. At 37D, a small amount of Prm2 was distributed in the seminiferous tubules, and the expression level of Bax decreased to 16.362 ± 3.267, indicating that apoptosis had begun to terminate. At 47D, 20 ± 4 spermatids were rediscovered in the seminiferous tubules, and the expression level of Bax was 10.176 ± 1.682, which was not different from the control level. This result indicates that 22AA can reversibly regulate spermatid apoptosis.

22AA affects occludin localization and distribution

Dual immunofluorescence was used to analyze the effect of 22AA on the localization and distribution of occludin. The multiplex immunohistochemical results for each group are shown in Fig. 1C. Occludin was mainly located at TJ between the basement membrane of seminiferous tubules and SCs in the control group. At 7D, a small amount of occludin was distributed at TJ between the basement membrane of seminiferous tubule and SCs. However, the total number of cells in seminiferous tubules was less than that in the control group. At 17D and 27D, no cells with blue nuclei were found in the seminiferous tubules. From day 17 to day 27, the expression of occludin gradually decreased. At 37D, occludin expression began to increase and was found to be distributed on the basement membrane of seminiferous tubules and SCs. At 47D, occludin expression and distribution and the morphological structure of seminiferous tubules were highly similar to those in the control group. These results suggest that 22AA can reversibly affect the location and distribution of occludin.

22AA downregulates occludin

To analyze the dynamic changes in the occludin expression level after TJ disintegration and reconstruction, total testicular proteins were extracted at 0, 7, 17, 27, 37, or 47 days for western blot analysis. Representative western blot results are shown in Fig. 2A. The relative occludin expression level was calculated using the occludin/ β-actin gray density ratio. The detailed values are shown in Fig. 2B. In the control group, the expression value was 0.9967. From 7D to 27D, the occludin expression level in the 22AA group gradually decreased to 0.1621, which was only 16.26% of that in the control group. Then, the expression level of occludin gradually increased to 0.3543 at 47D, which was approximately one-third the normal expression level (35.54%). Occludin expression was significantly different among the six groups (P < 0.05). The results showed that the expression level of occludin in the 22AA group at each time was significantly different from that in the control group (P < 0.01). These results indicate that 22AA can reversibly regulate occludin expression.

Figure 2 Effect of 22AA on the expression of occludin.

Representative western blot results are shown in A. The relative occludin expression level was calculated using the occludin/β-actin gray density ratio. The detailed values are shown in B. In the control group, the expression value was 0.9967. From 7D to 27D, the occludin expression level in the 22AA group gradually decreased to 0.1621, which was only 16.26% of that in the control group. Then, the expression level of occludin gradually increased to 0.3543 at 47D, which was approximately one-third the normal expression level (35.54%). Occludin expression was significantly different among the six groups (P < 0.05). The results showed that the expression level of occludin in the 22AA group at each time was significantly different from that in the control group. *P < 0.01.

Figure 3 Effect of artificially synthesized 22AA on the expression of miR-122-5.

The expression levels of miR-122-5p in each group are shown in A. The miR-122-5p expression level in the control group was 0.0408 and increased to 0.0539 at 7D. The miR-122-5p expression level in the 22AA group at 27D was the highest at 0.1293. Then, miR-122-5p expression gradually decreased to 0.0867 at 47D but was still higher than that in the control group. The results showed that the miR-122-5p expression levels were significantly different among the six groups (P < 0.01). The correlation between miR-122-5p and occludin expression in each group was analyzed using Pearson correlation coefficient. The linear relationship is shown in B. The results indicated that miR-122-5p and occludin expression are significantly negatively correlated (R2 =  − 0.4905, P < 0.01). *P < 0.01.

22AA upregulates miR-122-5p expression, and miR-122-5p expression is negatively correlated with occludin expression before 27D

To analyze the dynamic expression of miR-122-5p during TJ disintegration and reconstruction, total RNA in testes was extracted at 0, 7, 17, 27, 37, or 47 days. RT-PCR analysis was performed after the total RNA was reverse-transcribed into cDNA. The expression levels of miR-122-5p in each group are shown in Fig. 3A. The miR-122-5p expression level in the control group was 0.0408 and increased to 0.0539 at 7D. The miR-122-5p expression level in the 22AA group at 27D was the highest at 0.1293. Then, miR-122-5p expression gradually decreased to 0.0867 at 47D but was still higher than that in the control group. The results showed that the miR-122-5p expression levels were significantly different among the six groups (P < 0.01). The correlation between miR-122-5p and occludin expression in each group was analyzed using Pearson correlation coefficient. The linear relationship is shown in Fig. 3B. The results indicated that miR-122-5p and occludin expression are significantly negatively correlated (R2 =  − 0.4905, P < 0.01).

22AA changes the sperm count

To analyze the effect of 22AA on sperm count, the epididymis was extracted at 0, 7, 17, 27, 37, or 47 days. The sperm density was analyzed after the epididymis was shredded. The results are shown in Fig. 4A. The highest sperm density in the control group was 750.144 × 104/mL, after which it gradually decreased. The sperm density decreased to 164.278 × 104/mL at 27D and then gradually increased to 283.114 × 104 mL at 47D. The results showed a significant difference in sperm density among the six groups (P < 0.01).

Figure 4 Effects of 22AA on sperm density and litter size.

The sperm density was analyzed after the epididymis was shredded. The results are shown in A. The highest sperm density in the control group was 750. 144 × 104/mL, after which it gradually decreased. The sperm density decreased to 164. 278 × 104/mL at 27D and then gradually increased to 283.114 × 104 mL at 47D. The results showed a significant difference in sperm density among the six groups (P < 0.01). The number of embryos in the uterus on the 14th day of pregnancy was taken as a measure of fertility. The results are shown in B. The largest litter size in the control group was 12, and then, litter size decreased gradually to 3.667 at 27D. The litter size gradually increased to seven at 47D. The results showed that the difference in litter size among the six groups was significant. *P < 0.01.

22AA affects litter size

To analyze the effect of 22AA on the litter size of males, male mice were co-caged with female mice at 0, 7, 17, 27, 37, or 47 days. The number of embryos in the uterus on the 14th day of pregnancy was taken as a measure of fertility. The results are shown in Fig. 4B. The largest litter size in the control group was 12, and then, litter size decreased gradually to 3.667 at 27D. The litter size gradually increased to seven at 47D. The results showed that the difference in litter size among the six groups was significant (P < 0.01).

Discussion

As a component of TJ, occludin is the structural basis for TJ formation between SCs in the seminiferous epithelium. The programmed opening/resealing of TJ ensures normal progression of spermatogenesis, and abnormal opening/resealing can affect the normal spermatogenesis process (Langbein et al., 2003). Interference with the functional status of occludin protein in testicular SCs can result in infertility. TJ of the BTB are different from TJ of the blood–brain barrier and other barriers, and the specific function of BTB TJ between SCs is related to spermatogonia cell activity and differentiation. The disintegration and reconstruction of TJ between SCs is an important process (Malminen et al., 2003).

SCs are the structural basis of TJ in the testis (D’Aurora et al., 2015). The main functions of SCs include providing structural support, creating the BTB, participating in germ cell movement and ejaculation, and nurturing germ cells through the secretion process (Tian et al., 2014; Rebourcet et al., 2016; Nagai et al., 2016; Nicholls et al., 2012). In the present study, to analyze the dynamic changes in SCs during the process of TJ disintegration and reconstruction, a 22AA-induced TJ destruction animal model was utilized, and an immuno-double-labeling technique was used for analysis. WTI was employed as an SC marker (Wang et al., 2019). The results showed that the number of cells and the seminiferous tubule wall thickness were decreased in SCs at 7D compared with the control group. These results were consistent with those reported by Chung et al. (2001). No blue nuclei were found in the seminiferous tubules at 17D or 27D, and the intercellular boundary had completely disappeared. WT1 was distributed throughout the seminiferous tubules. At 37D, a few SCs began to be found in the basal layer of the seminiferous tubules, and spermatogenesis began to recover. At 47D, there was almost no difference in SCs between the 22AA group and the control group, indicating that spermatogenesis had returned to normal. These phenomena reveal for the first time the changing pattern of SCs in the process of TJ disintegration and reconstruction. Next, dynamic changes in spermatids were analyzed using Prm2 as the spermatid marker (Zalata et al., 2016; Kleene & Bagarova, 2008). The results of multiplex immunohistochemistry showed that there were 55 ± 5 spermatids in the seminiferous tubules of the control group. At 7D, the spermatid count decreased to 15 ± 3, suggesting that the spermatids gradually became apoptotic. No spermatids were found in seminiferous tubules at 17D or 27D. This finding indicates that the spermatids were completely apoptotic. A small amount of Prm2 was distributed in the seminiferous tubules at 37D, suggesting that spermatogenesis had begun to recover. At 47D, recovery of spermatids in seminiferous tubules was visible, and the spermatids numbered 20 ± 4, which was consistent with the experimental results of Chung et al. (2001). This finding indicates that spermatogenesis had returned to normal. These data suggest that 22AA mainly reduced the number of spermatids in seminiferous tubules before 27D. After 27D, the normal structure of the TJ was restored, and spermatogenesis resumed.

The TJ-related structural proteins include ZO-1, ZO-2, and multiple claudin genes. Wong et al. found that the 44-amino-acid peptide in the second extracellular loop of occludin had no effect on the expression level of ZO-1, ZO-2, or cingulin in the Xenopus kidney epithelial cell line A6 (Wong & Gumbiner, 1997). Therefore, in the present study, only the dynamic expression of occludin was quantitatively analyzed when investigating the mechanism underlying the destruction and recovery of TJ induced by 22AA. The expression level of occludin in the 22AA group gradually decreased from 7D to 27D until reaching only 14% of that in the control group. Then, the expression level of occludin gradually increased. At 47D, occludin expression recovered to approximately one-third the normal expression. These results suggest that 22AA can reduce the expression level of occludin at the disintegration stage of TJ. Therefore, it can be concluded that 22AA can downregulate occludin, leading to disintegration of TJ. The 22AA-induced disruption in the TJ barrier is possibly mediated by one of the following mechanisms. First, it might be possible that SCs were using these peptides as building blocks for TJ assembly. However, because they did not have the structural confirmation of the entire molecule to reinforce the TJ functionality, TJs became perturbed and disrupted. Second, homotypic interactions of the synthetic peptides with other intact occludin molecules between two neighboring SCs caused the recruitment of intact and functional occludin to the same site to become impossible. Thus, the TJ permeability barrier became disrupted.

There are three possible reasons for the decrease in protein expression. The first reason is protein degradation (Majolee, Kovacevic & Hordijk, 2019). Occludin phosphorylation and ubiquitination regulate TJ (Murakami, Felinski & Antonetti, 2009). The western blot results showed that the molecular weight of occludin was consistent with the actual size, suggesting that occludin did not degrade. This result indicates that 22AA did not cause occludin ubiquitination. The second potential reason is cell apoptosis because cell apoptosis prevents cells from expressing relevant proteins (Schiffmann et al., 2019). Therefore, SC and spermatid apoptosis might lead to a decrease in occludin expression. The third reason could be inhibition of transcription or translation by noncoding RNA (Feng et al., 2019). To investigate the occludin downregulation mechanism, we analyzed the expression of several microRNAs (miRNAs) that can target occludin (data not shown), among which only the expression of miR-122-5p was associated with occludin expression. miR-122-5p is encoded on chromosome 18q21.31 and is derived from the hcr gene transcript. miR-122-5p plays an important role in cell cycle regulation, cell proliferation and cell apoptosis (Wang & Wang, 2019) and is associated with multiple diseases (Wang & Wang, 2019; Wen et al., 2018; Jiang et al., 2017; Cortez-Dias et al., 2016). Previously, we analyzed the correlation between miR-122-5p and occludin protein, and the results showed that miR-122-5p was negatively correlated with occludin expression (Zhu et al., 2019). Our other recent results showed that miR-122-5p regulates occludin expression through the AACACTCCA sequence of the occludin 3′UTR, thereby regulating the formation and tightness of TJ between SCs (submitted). We further employed real-time RT-PCR to assess whether 22AA affects miR-122-5p expression. The current results showed that the expression level of miR-122-5p gradually increased from 0 to 27D and then began to decrease. The change in the miR-122-5p expression level was opposite that of occludin. Correlation analysis showed a significant negative correlation between miR-122-5p and occludin expression from 0 to 47D (Fig. 3B). These results indicate that 22AA increased the expression of miR-122-5p, which mediated downregulation of occludin expression, thereby causing disintegration of TJ. However, the mechanism by which 22AA regulates mir-122-5p expression remains to be further studied.

To investigate the causes of SC and spermatid dysfunction, Bax was used as an apoptosis marker protein to analyze cell apoptosis in seminiferous tubules. Bax, belonging to the Bcl-2 gene family, is the most important apoptotic gene in humans. The encoded Bax protein forms heterodimers with Bcl-2 and has an inhibitory effect on Bcl-2. Bax is one of the most important apoptosis-promoting genes (Szymona et al., 2019). Bax expression is also closely related to spermatogenesis (Russell et al., 2002). In the present study, Bax expression was found in the seminiferous tubules of the control group at a low expression level of 11.373 ± 10.532. Yan et al. (2000) also found that Bax was expressed in various types of cells in normal testicular tissues. At 7D, the expression level of Bax was 16.783 ± 10.157, which was higher than that in the control group, indicating that the cells in seminiferous tubules began to undergo apoptosis. However, at this stage, the structure of SCs and spermatids in the tubules was relatively intact. The expression level of Bax increased to 20.521 ± 5.781 and 30.253 ± 12.274 at 17D and 27D, respectively. At this stage, the tubules were filled with a large amount of red Bax, no blue nuclei were observed, and the intercellular boundary had completely disappeared, indicating that all the cells in the seminiferous tubules had already undergone apoptosis. At 37D, the expression level of Bax was slightly lower than at 27D. In the basal layer of the seminiferous tubules, blue nuclei began to appear, indicating that apoptosis had slowed and the spermatids and SCs of the seminiferous tubules had started to recover. At 47D, the expression level of Bax was not significantly different from that in the control group, and the structure and number of spermatids and SCs in seminiferous tubules were not different from those in the control group, indicating that spermatogenesis had fully recovered. These results suggest that 22AA can induce apoptosis in seminiferous tubules before 27D.

To analyze the effect of 22AA on sperm count, the epididymis was extracted at 0, 7, 17, 27, 37, or 47 days. Sperm density was analyzed after the epididymis was shredded. The results are shown in Fig. 4A. The highest sperm density in the control group was 750.114 × 104/mL, and then, the density gradually decreased, dropping to 164.278 × 104/mL at 27D; subsequently, the sperm density gradually increased to 283.114 × 104 mL at 47D. However, compared with the control group, the sperm density was greatly decreased. The decrease in sperm count may be due to the following reasons. First, high expression of Bax promotes the apoptosis of type A spermatogonial stem cells (Russell et al., 2002), thereby reducing spermatogenesis. Second, after the BTB is destroyed, immune cells enter the seminiferous tubules and engulf many sperm (Adegoke et al., 2018). Third, sperm undergo apoptosis or autophagy in the epididymis (Grunewald, Fitzl & Springsguth, 2017; Sinkakarimi, Solgi & Colagar, 2020). Further analysis is needed to determine which reason explains the decrease in sperm count. To analyze the effect of 22AA on litter size, male mice were co-caged with female mice at 0, 7, 17, 27, 37, or 47 days. The number of embryos in the uterus on the 14th day of pregnancy was used to measure fertility. The highest litter size in the control group was 12, but litter size decreased gradually to seven at 27D. Afterwards, the litter size gradually increased to 6.67 at 47D.

In summary, this study investigated the effect of 22AA on TJ by using a 22AA-induced TJ destruction animal model. The results showed that before 27D, 22AA promoted SC and spermatid apoptosis, downregulated occludin, upregulated miR-122-5p, and decreased sperm density and litter size. After 27D, the occludin expression increased, miR-122-5p expression decreased, both sperm density and litter size rebounded, cell apoptosis stopped, and spermatogenesis began to recover. Therefore, it can be concluded that 22AA destroys TJ by downregulating occludin and inducing cell apoptosis. After 27D, TJ and spermatogenesis functions return to normal.

Supplemental Information

Supplemental Information 1 Raw data and figures

Click here for additional data file.

Additional Information and Declarations

Competing Interests

Author Contributions

Animal Ethics

Data Availability

The authors declare there are no competing interests.

Maoying Zhu and Lumin Fei conceived and designed the experiments, analyzed the data, authored or reviewed drafts of the paper, and approved the final draft.

Juan Lu conceived and designed the experiments, performed the experiments, prepared figures and/or tables, authored or reviewed drafts of the paper, and approved the final draft.

Jianyun Shen performed the experiments, prepared figures and/or tables, and approved the final draft.

Deyu Chen conceived and designed the experiments, performed the experiments, analyzed the data, prepared figures and/or tables, and approved the final draft.

The following information was supplied relating to ethical approvals (i.e., approving body and any reference numbers):

The experiment was approved by the Experimental Animal Ethics Committee of Fuyang Normal University, China (Grant No. 20200006).

The following information was supplied regarding data availability:

The raw measurements are available in the Supplemental Files.

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
