# Peer review of "A 22-amino-acid peptide regulates tight junctions through occludin and cell apoptosis"

_PeerJ, doi:10.7717/peerj.10147_

## Round 0.1 · original submission · Major Revisions

Dear Dr. Zhu and colleagues:

This is a resubmission of a previously submitted manuscript which I handled. The previous reviewers were not invited, as were some new reviewers.

Thanks for submitting your manuscript to PeerJ. I have now received three independent reviews of your work, and as you will see, the reviewers raised some concerns about the research. Despite this, these reviewers are optimistic about your work and the potential impact it will have on research studying occludin and tight junctions regulation. Thus, I encourage you to revise your manuscript, accordingly, taking into account all of the concerns raised by both reviewers.

While the concerns of the reviewers are relatively minor, this is a major revision to ensure that the original reviewers have a chance to evaluate your responses to their concerns.

There are many suggestions, which I am sure will greatly improve your manuscript once addressed.

Reviewer 2 has recommended some minor experiments; either conduct these or state why this is not necessary.

Importantly, please ensure that an English expert has edited your revised manuscript for content and clarity. Please also ensure that your figures and tables contain all of the information that is necessary to support your findings and observations. Please also make sure that all typos are corrected.

I look forward to seeing your revision, and thanks again for submitting your work to PeerJ.

Good luck with your revision,

-joe

Reviewer 1 ·

Basic reporting

Some more background introduction of occludin will be nice, especially known mechanisms of occludin in cell apoptosis and human fertility.

Experimental design

In statistical analysis of methods, the significant difference level was set as P<0.05. However, in every figure legend, the significant difference level was set as P<0.01. Please correct it.

Validity of the findings

In discussion part, authors explained “The western blot images in this study showed only one band for occludin, and thus, no degradation was found”. This is a completely wrong conclusion. Protein degradation could be affected by many ways, e.g., ubiquitination, SUMOylating, autophagy-related protein degradation, etc. Authors should re-write this sentence or do protein degradation assay to exclude this possibility.
Similarly, in discussion part, author stated “Therefore, SC and spermatid apoptosis might lead to a decrease in occludin expression.” However, at 47D, spermatogenesis had returned to normal, while expression level of occludin was approximately one-third the normal expression level. Why did the expression of occludin not return to normal? Authors should explain this.

Additional comments

In the paper, authors investigated the mechanism of 22-amino-acid peptide in regulating tight junctions through occludin and cell apoptosis. The study gives an interesting new prospective of a mechanism not deeply studied for male fertility. The topic presented is of interest and the data are quite important; however, the manuscript in this form presents several problems.
1. In figure 2A, why did authors use one β-actin for two different groups?
2. From figure 2 to figure3, it is too weird. Authors analyzed the expression of several microRNAs (miRNAs) that can target occluding, but did not show the results. I think that reporting this data is fundamental.
3. Could miR-122-5p target other structural proteins of TJ?

·

Basic reporting

The authors present the results that provide experimental support for the role of 22AA peptide in SC and spermatid apoptosis and in sperm density regulation. The figures are relevant but need some modifications that is mentioned in later section. I commend the authors for doing a good literature review with a well written introduction section. The experiments are mostly well executed with control; the exceptions are mentioned in general comments. Some minor additional experiments can be done to improve the overall quality of the paper.

Experimental design

The study is of importance and within the scope of Peer J. The research plan is well defined and relevant experiments are performed. Some modifications and addition of better picture can improve the clarity of the paper. Additionally, some minor experiments can be done to improve the overall quality of the paper to meet the standard of Peer J.

Validity of the findings

The study is fairly robust and looks statistically sound. Conclusions are well supported by results except the ones that I have mentioned in the general comments.

Additional comments

This is an interesting study and the strength of the study is the use of in-vivo model system. The manuscript is concise and clear. Although, there are certain topics that need attention as followed.

1. The abstract claims (line 7-8) that distribution of occluding was analyzed by immunofluorescence electron microscopy but I do not see any such data in the manuscript.

2. Fig 1 is a very important piece of evidence but there is some ambiguity in the data. The authors state that they counted sertoli cells and spermatids from this imaging data but the picture shows no specific staining that can be used to differentiate sertoli cells from spermatids. I only see blue staining (nucleus) and red (Bax). I see in the method section, the authors mentioned WT1 and Prm2 immunostaining but no such staining is clearly visible in these pictures (Fig 1). They should use images with better quality and contrast. I would also suggest to add different panels of single stain and then last one with merged. Additionally, The picture shows a single seminiferous tubule, I would recommend to add pictures with multiple seminiferous tubules in one field for each condition.
I would also suggest adding biochemical data to consolidate this imaging result. One experiment that can be performed to measure functional outcome of loss of sertoli cells is to measure Anti-müllerian hormone (AMH) level. AMH is produced by sertoli cells in early developmental stage. If there is loss of sertoli cells, a drop in AMH level would be expected. Considering the importance of this figure, the above mentioned points need to be addressed.

3. For Fig 2 –Fig 4, the figure sub labeling (A/B) can be placed at the left top of each figs.

4. Fig 2 shows quantification of WB result; the details of quantification methods including the software used need to be given in the method section.

5. Fig 3: This result is not really convincing. I see an increase in expression of miR-122-5p from 0 to 0.1! Is this minor increase physiologically relevant? This result can be excluded. The increase shown here is very marginal and the result is not adding any substance to the overall conclusion of the paper.

6. The results show that the effect of 22AA disappeared post 27D but the authors have not given much details of this incongruity in the discussion section. One line is written saying that the degradation of 22AA happens post 27D but I don’t see any data in the manuscript supporting this hypothesis. If any data exists, it should be included or else if the comment is based on previous observation that should be mentioned.

7. In table 1: The table is labeled as ‘Antibody types and sources’ but this table also enlists other reagents like PBS, so I would suggest labeling it as- Table of resources/Source of reagents.

8. Fig 1 legend: Consider changing the title of the legend to make it more objective.

9. There are typos that need to be corrected, some examples are as followed:
Line 42: remove ‘.’ before TJ
Line 60: at the end of the line ‘??’ is there. One’?’ to be removed.
Line 179: add a ‘-’ in t test.

Reviewer 3 ·

Basic reporting

The study investigates a relevant research question to understand the mechanisms underlying the regulation of tight junctions by 22AA. How 22AA affects the expression and localization of occludin? The manuscript covers appropriate content (literature, description of methodology and results) in specific sections.

Experimental design

Experimental design is well described now.

Validity of the findings

Findings appear to be valid.

Additional comments

Language errors need correction:
In vivo and in vitro should be mentioned in italic font throughout the text.
L33: adluminal
L34: lumen
L42: punctuation error
L314-319, L353-364, L369-373: These lines describe and mention the results which in my view is Repetition of results, it would be more useful to precisely discuss the finding here and further reduce the extensive text in discussion.

---

## Round 0.2 · Minor Revisions

Dear Dr. Zhu and colleagues:

Thanks for revising your manuscript. The reviewers are very satisfied with your revision (as am I). Great! However, there are a few concerns to address. Please attend to these issues ASAP so we may move towards acceptance of your work.

Best,

-joe

Reviewer 1 ·

Basic reporting

no comment

Experimental design

no comment

Validity of the findings

no comment

Additional comments

Thank you very much for your response to my concerns. Well done!

·

Basic reporting

It looks much better than the previous version. Few minor changes can improve the manuscript further.

Experimental design

Well designed

Validity of the findings

No comments

Additional comments

1. I would suggest adding some of the short and long term medical consequences of the 22AA mediated effects.
2. Fig 1: Images need to be properly labeled. I would specifically suggest labeling control and 22AA peptide treatment panels.
3. The author should discuss in detail the reason of disappearance of the 22AA mediated phenotypes post 27 days.

---

## Round 0.3 · accepted · Accept

Dear Dr. Zhu and colleagues:

Thanks for revising your manuscript based on the concerns raised by the reviewers. I now believe that your manuscript is suitable for publication. Congratulations! I look forward to seeing this work in print, and I anticipate it being an important resource for groups studying occludin and tight junctions regulation. Thanks again for choosing PeerJ to publish such important work.

Best,

-joe